# Human Papillomavirus (HPV) Vaccine Coverage and Confidence in Italy: A Nationwide Cross-Sectional Study, the OBVIOUS Project

**DOI:** 10.3390/vaccines12020187

**Published:** 2024-02-12

**Authors:** Marco Montalti, Aurelia Salussolia, Angelo Capodici, Francesca Scognamiglio, Zeno Di Valerio, Giusy La Fauci, Giorgia Soldà, Maria Pia Fantini, Anna Odone, Claudio Costantino, Julie Leask, Heidi J. Larson, Jacopo Lenzi, Davide Gori

**Affiliations:** 1Department of Biomedical and Neuromotor Science, Alma Mater Studiorum—University of Bologna, 40126 Bologna, Italydavide.gori4@unibo.it (D.G.); 2Interdisciplinary Research Center for Health Science, Sant’Anna School of Advanced Studies, 56127 Pisa, Italy; 3Department of Public Health, Experimental and Forensic Medicine, University of Pavia, 27100 Pavia, Italy; 4Department of Health Promotion Sciences, Maternal and Infant Care, Internal Medicine and Excellence Specialties “G. D’Alessandro”, University of Palermo, 90133 Palermo, Italy; 5School of Public Health, University of Sydney, Sydney, NSW 2006, Australia; 6Infectious Disease Epidemiology, London School of Hygiene & Tropical Medicine Institute of Health Metrics, London WC1E 7HT, UK

**Keywords:** Human Papillomavirus (HPV), vaccines, vaccination, prevention, vaccine coverage, vaccine confidence, behavioral and social determinants, vaccine hesitancy

## Abstract

Background: Human Papillomavirus (HPV) vaccination rates are still below the target due to vaccine refusal or delay, lack of knowledge, and logistical challenges. Understanding these barriers is crucial for developing strategies to improve HPV vaccination rates. Methods: This cross-sectional study used a questionnaire to investigate social and behavioral factors influencing decision making about the HPV vaccine. The survey was conducted from 11 April to 29 May 2022 and involved 10,000 Italian citizens aged ≥ 18 years. The sample was stratified based on region of residence, gender, and age group. Results: 3160 participants were surveyed about themselves, while 1266 respondents were surveyed about their children’s vaccine uptake. Among females aged ≥ 26 years, the national average HPV vaccine uptake was 21.7%, with variations across different regions. In the 18–25 age group, females had a vaccine uptake (80.8%) twice as much as males (38.1%), while vaccine uptake among male and female children aged 9–11 was similar. Conclusions: The OBVIOUS study in Italy reveals factors influencing low HPV vaccine uptake, suggesting targeted approaches, tailored information campaigns, heightened awareness of eligibility, promoting early vaccination, addressing low-risk perception among males, addressing safety concerns, and enhancing perceived accessibility to improve vaccine uptake and mitigate health risks.

## 1. Introduction

There are over 450 different types of Human Papillomaviruses (HPVs) that belong to the Papillomaviridae family [1]. The most common genotypes associated with causing genital warts, which is a benign condition affecting the external genitalia, are #6 and #11. These genotypes are considered as “low-risk” within the spectrum of HPVs. On the other hand, the “high-risk” group comprises at least 13 HPV genotypes that are known to cause various cancers, particularly cervical cancer, as well as anogenital and head and neck cancers. Among these high-risk genotypes, HPVs #16 and #18 are responsible for 70% of cervical cancer cases worldwide [2].

Within the European Union (EU), cervical cancer is the second most prevalent cancer following breast cancer among women aged 15 to 44. Each year, around 33,000 cases of cervical cancer are reported in the EU, leading to approximately 15,000 fatalities [3]. Italy experiences approximately 3500 new cases of cervical cancer and 1500 deaths annually [4]. 

Due to its high transmissibility, widespread presence, and significant long-term implications, HPV poses a significant public health challenge, making it an ideal candidate for screening programs and comprehensive vaccination campaigns. Currently, three preventive HPV vaccines have been authorized for use. These include a bivalent vaccine targeting HPV16/18 virus-like particles (VLPs), a quadrivalent vaccine targeting HPV 6/11/16/18 VLPs, and the latest nonavalent vaccine targeting HPV 6/11/16/18/31/33/45/52/58 VLPs [5]. Clinical trials and post-marketing studies have shown that all the vaccines have exhibited a favorable efficacy and safety profile [6,7,8,9]. Specifically, the nonavalent vaccine provides protection against the serotypes responsible for 90% of cervical cancer cases as well as cases of genital warts [5]. The World Health Organization (WHO) recognizes the cost-effectiveness of HPV vaccination and its crucial role as a public health measure in preventing HPV-related diseases. As a result, the WHO has prioritized the inclusion of HPV vaccination in national immunization programs [2]. 

In Italy, HPV vaccination has been offered actively and free of charge to girls in their twelfth year of life (eleven years old) since 2007/2008. As gender-neutral HPV immunization campaigns have proven to be more resilient than those targeting only female adolescents [10,11,12], three Italian regions (Sicily, Apulia, and Molise) introduced, as early as 2015, HPV vaccination also for 11-year-old males; the whole country reached full operation in actively offering the vaccination to males in 2018 [13]. Catch-up vaccination is recommended for women at least up to the age of 26, utilizing the opportunity of the first screening for cervical cancer prevention. For men, catch-up vaccination is recommended at least up to the age of 18, in cases where they have not been previously vaccinated or have not completed the vaccination cycle. According to the scientific literature [14], certain regions of Italy such as Emilia-Romagna and Friuli-Venezia Giulia have implemented extended measures to provide free-of-charge HPV vaccination and actively reach out to individuals living with HIV [13]. In recognition of the higher risk of HPV infections and associated lesions among specific populations such as males who have sex with males and sex workers, Emilia-Romagna made the decision in 2019 to extend the offer of free-of-charge HPV vaccination to these subgroups [15,16,17]. In addition to the specific populations mentioned earlier, women who have undergone treatment for high-grade cervical neoplasms are also considered at higher risk. Vaccinating these women after cervical surgery can help reduce the risk of recurrence and minimize the adverse effects associated with repeated surgery. Furthermore, HPV vaccination can help prevent new infections caused by the HPV genotypes covered by the vaccine. By including this group in vaccination efforts, the aim is to provide additional protection and support their long-term health outcomes [18,19,20]. As of now, the free offer for HPV vaccination to women with previous HPV lesions is available in 15 regions in Italy. Lastly, most Italian regions provide subsidized payment for other age groups that are not included in active calls or catch-up activities.

Despite the active call and free-of-charge offer, HPV immunization campaigns in Italy are facing challenges in achieving high coverage rates, which are still below the target set by the National Vaccination Prevention Plan of 95% for 12-year-olds. Data from the Italian Ministry of Health reveals that, as of 2020, 62.2% of females and 49.9% of males born in 2006 have received the HPV vaccine. Additionally, the uptake of HPV vaccination has declined in the past two years due to the obstacles presented by the COVID-19 pandemic [13]. 

Based on these data, characterizing factors hindering the achievement of adequate coverage is of the utmost importance. Vaccine uptake is affected by a range of factors. These can relate to beliefs and feelings, social influences, and practical and logistical issues. Hesitancy, which the WHO has re-defined as a motivational state of being conflicted or opposed to vaccination, is a major issue in Italy [21,22].

In a study by Donati et al., an HPV vaccine uptake of 64% was found among women aged 18–26 years, and risk perception, cost of the vaccine, and advice from healthcare providers were cited as the main factors in deciding whether to vaccinate [23]. Another study by Di Giuseppe et al. found extremely low knowledge about HPV infection in a sample of Italian adolescents and young women [24]. Several studies have investigated the factors influencing parents’ decisions regarding the vaccination of their children, particularly in relation to HPV. These studies have highlighted various barriers to HPV vaccination, including fear of adverse events, lack of confidence in a new vaccine, receiving conflicting information from healthcare providers, and limited availability of information about HPV vaccination. These factors contribute to vaccine hesitancy and play a significant role in influencing parental decisions regarding HPV vaccination for their children [25,26].

Indeed, understanding the complete spectrum of factors and barriers to HPV vaccination is crucial in developing effective policy and programmatic solutions. To address this need, the objective of the study was to identify the primary social and behavioral factors that impede HPV vaccination among individuals who have been recommended to receive the vaccine. The findings from this study will provide valuable insights for national, regional, and local program planning and policy development, enabling targeted strategies to overcome barriers and improve HPV vaccination rates.

## 2. Materials and Methods

### 2.1. Study Design and Data Collection

The study was conducted using a cross-sectional, computer-assisted web interviewing (CAWI) questionnaire. The data collection took place from 11 April to 29 May 2022, with the assistance of the professional online provider Dynata (accessed on 8 February 2024; https://www.dynata.com/). A sample of 10,000 Italian citizens aged 18 years and above was surveyed. The sampling approach involved stratification based on proportionate allocation, considering the first-level NUTS (Nomenclature of Territorial Units for Statistics) statistical regions of residence, gender, and age groups. The age groups included were 18–24, 25–34, 35–44, 45–54, 55–64, and 65 years and above. 

The survey was structured to be completed within approximately 10 min and consisted of seven sections. The first section focused on demographics and living conditions, gathering information about participants’ educational background. Sections #2 to #5 were dedicated to collecting data specifically related to vaccination against HPV (which was the main focus of the study), as well as vaccinations against pneumococcus, varicella-zoster virus, rotavirus, and influenza. The final section of the survey, not subject to analysis in this study, explored participants’ political orientation, attitudes toward SARS-CoV-2 vaccination, perceptions of science, and views on alternative medicine.

To align with the vaccination recommendations set by the Italian Ministry of Health, the sample for the HPV section of the survey targeted specific age groups. For women, the focus was on those between 18 and 59 years of age, while for men, the target group was between 18 and 25 years of age. Parents or guardians of offspring aged 9 to 17 years were also included in the sample. The year of birth was used to differentiate respondents who fell within the target groups. This approach ensured that the survey captured data from individuals who were directly relevant to the HPV vaccination recommendations and programs in Italy. Additionally, it accounted for adult males and women over the age of 25 who may not be eligible for free vaccination but could still choose to get vaccinated due to their close association with the target population.

Data management by Dynata for the survey was conducted in compliance with the General Data Protection Regulation (GDPR) of the EU. The GDPR ensures the protection and privacy of personal data of individuals within the EU. Furthermore, the survey adhered to all relevant requirements outlined by Italian regulations, ensuring that data collection, storage, and processing were conducted in accordance with applicable laws and guidelines in Italy. These measures were implemented to safeguard the privacy and confidentiality of participants’ information throughout the survey process.

### 2.2. Questionnaire

Prior to the full implementation of the survey, cognitive testing was conducted to assess its effectiveness and gather feedback. The testing phase, conducted with 100 respondents, was used to make revisions and improvements to the questionnaire. The first section of the survey was based on the WHO Behavioural and Social Drivers (BeSD) survey, which provided a framework for investigating the behavioral and social factors influencing vaccination decisions [21]. 

The survey included a total of 21 questions covering various demographic and relevant factors. These questions covered information such as gender; date of birth; region and province of residence; educational attainment; occupation; living arrangement; ability to pay necessary living expenses; pregnancy in late 2021 (only for women); daily living difficulties due to physical, psychological, or sensory impairments; weight; height; presence of chronic respiratory diseases; cardiovascular diseases; and diabetes. Additionally, questions were posed regarding the location where the majority of vaccinations took place, the preferred location to receive the vaccination, views of friends and family on vaccination, having children, gender and date of birth of the youngest offspring, and agreement on vaccination decisions between partners. Based on the responses to these questions, targeted respondents were directed to the section of the survey specifically dedicated to the HPV vaccine, ensuring relevant and targeted data collection for this specific topic.

Vaccine-specific sections were developed following the domains of the WHO BeSD framework [21]: thinking and feeling, social processes, motivation, practical issues, and vaccination. Questions in the HPV section were as follows: being vaccinated for HPV (“yes” or “no”); worry about catching the HPV infection (four response options: “not worried”, “a little worried”, “quite worried”, and ”very worried”); vaccine safety perception (four response options: ”very safe”, “quite safe”, “quite unsafe”, and “very unsafe”); awareness of being in the target population for the vaccination program (three response options: “yes”, “no”, and “I don’t know”); ease in accessing HPV vaccination (four response options: “very easy”, “quite easy”, “quite difficult”, and “very difficult”); affordability of the HPV vaccination (four response options: “not at all affordable”, “not very affordable”, “quite affordable”, and “very affordable”); and payment or not for taken vaccination (two response options: “free of charge” and “paid”). Those who answered that they had not been vaccinated for HPV were asked about their current intention to get the vaccine (“yes” or “no”). Parents or guardians were specifically instructed to answer all the questions on behalf of their offspring, even if they themselves met the criteria for the adult population. The questionnaire tool translated into English can be found in Appendix A. Back translation was performed to check for the accuracy and soundness of the original Italian-to-English translation. 

### 2.3. Statistical Analysis

Post-stratification by gender, age group, and area of residence revealed that non-response in certain segments of Italy’s population was minimal and did not impact the study’s estimates based on the overall sample of 10,000 [27]. Consequently, adjusting sampling weights for the targeted subset of respondents for HPV vaccination was deemed unnecessary. No formal power analysis was conducted prior to data collection. Based on resource availability and the aim of collecting enough data to provide reasonably precise estimates originating from each section of the questionnaire, including further stratified analyses, we determined a minimum size of 10,000 for the entire OBVIOUS sample.

All variables were stratified by NUTS and by age and gender groups (females aged ≥ 26 years, males vs. females aged 18–25 years, male vs. female offspring aged 9–11 years, and male vs. female offspring aged 12–17 years). Data were visualized with the aid of thematic maps with pie charts and with the aid of square charts. Square charts, also called waffle charts, are a form of pie chart that use 10 × 10 grids instead of circles to represent percentages.

Multivariable multinomial logistic regression analysis was conducted to investigate the factors influencing a three-category nominal outcome consisting of three mutually exclusive response options, i.e., “I did get the vaccine” vs. “I did not get the vaccine, but I would” vs. “I do not want to get vaccinated”. We opted to use a single multinomial logistic model in place of a series of binary logistic models to ensure estimator efficiency and consistency across the study outcomes. In keeping with the vaccination framework outlined by the BeSD Expert Working Group [21], the regression model incorporated the following covariates as potential determinants of vaccine uptake, delay, and refusal: attitudes and beliefs regarding HPV infection and vaccination (perceived worry and safety concerns), social processes (friends’ and family’s opinions on vaccination, and gender), and practical considerations (awareness of higher priority for vaccination, perceived ease of access to healthcare for vaccination, and perceived affordability of vaccines). Relevant sociodemographic determinants (age group, statistical region of residence, place of residence, level of urbanization, and educational attainment) were also considered. Year of birth was used to discriminate between right and wrong respondents by saying they had higher priority for vaccination. We could not measure the concept of “motivation” alone, since intentions were combined with behavior (vaccination) in the outcome variable. Therefore, it could not be tested as a mediator in the association of beliefs and social processes with vaccine uptake.

The effect of covariates was assessed by examining the marginal effect of changing their values on the average predicted probability of observing each outcome. The marginal effect was computed as a discrete difference in probabilities (Δ), with 95% confidence intervals (CIs) obtained using the delta method. Covariate categories occurring in <5% of the sample were combined with adjacent lower or upper classes to improve the stability and efficiency of regression estimates. The Small–Hsiao test of independence of irrelevant alternatives (IIA) did not indicate the need for alternative model specifications in which binary logit coefficients do not converge in probability to the same values as the multinomial logit coefficients, such as the nested logit model. Lastly, to check for the presence of moderators, that is, covariates Z that change the effect of other independent variables X on the uptake, pairwise interaction terms Z × X were included in the model one at a time and their statistical significance was tested using the likelihood-ratio (LR) test. To control for type I errors related to multiple testing, the significance level for interactions was set at 0.001.

All analyses were performed using Stata software, version 17 [28], and were conducted separately on individuals responding on their own behalf and those responding on behalf of their offspring. No multicollinearity issues were found in regression analysis, which means that the variance inflation factor was <5 and the condition index was <10 for each covariate.

## 3. Results

### 3.1. Demographic Information of the Adults

Table 1 presents a summary of the sample characteristics. Out of the total respondents, 3608 (36.1%) were specifically queried about their personal HPV vaccine uptake. Following the exclusion of 448 participants who were unable to recall their vaccination status, the analysis encompassed a sample size of 3160 (31.6%). Gender distribution revealed that males constituted 14.5% of the sample, while females accounted for 85.1%; non-binary individuals and those who preferred not to disclose their gender comprised 0.4% of the sample. The age group most represented in the study was individuals aged 18–25 years, making up 33.3% of the sample. Most participants (47.8%) resided in towns or suburban areas characterized by intermediate population density. Educational attainment indicated that 60.2% of the sample held a high school diploma, while 30.0% possessed a university degree or higher. Cohabitating with a partner was reported by 43.1% of respondents, and 39.2% were living with parents or other family members. Concerning economic circumstances, 59.6% of participants reported facing moderate or significant difficulties in meeting their basic life necessities with their available resources.

Appendix A provides insights into the vaccination preferences and attitudes within the sample. The data reveal that most respondents (64.5%) primarily obtained vaccines from Vaccine Hubs, and a large proportion (37.6%) expressed a preference for receiving vaccines at such hubs. Furthermore, the percentage of respondents with unfavorable views from friends and family towards vaccination was relatively low, with only 3.0% holding unfavorable views and 3.8% expressing very unfavorable views.

### 3.2. Demographic Information of the Offspring

Table 2 presents an overview of the demographic information concerning respondents who provided details about their youngest offspring’s HPV vaccine. The analysis included a total of 1266 respondents, with 581 (45.9%) being males, 684 (54.0%) being females, and one (0.1%) not disclosing their gender identity. One hundred nineteen participants were excluded from the analysis due to their inability to recall their youngest offspring’s vaccination status. Regarding the decision-making process for vaccination, the data revealed that only in 39.0% of cases the decision was made jointly between partners. 

In terms of the age range of the youngest offspring, Table 3 indicates that 66.3% of the respondents reported their offspring’s age as being between 12 and 17 years, while 33.7% fell between 9 and 11 years of age. 

It is worth noting that the data presented in this section can be further explored with regards to gender and geographical distribution within Italy.

### 3.3. Vaccine Uptake by Geography

The vaccine uptake in our sample varied according to geographical area, gender, and age group. Figure 1, Figure 2, Figure 3 and Figure 4 provides a visual representation of these variations.

In females aged 26 years and older (Figure 1), the average vaccine uptake at the national level was 21.7%. The highest uptake was observed in the northeast with 27.4%, while the lowest uptake was recorded in the center with 18.4%. 

Analyzing the vaccine uptake by gender in the 18–25 age group (Figure 2), we found that the average uptake nationwide was 80.8% in females and 38.1% in males. The northeast again had the highest values, with 85.4% uptake in females and 47.4% uptake in males. Conversely, the islands had the lowest values, with 74.1% uptake in females and 32.6% uptake in males. At the national level, 13.6% of females aged 18 to 25 were unaware of being the target of HPV vaccination, compared to 38.7% of males in the same age group.

Moving on to the vaccine uptake among respondents’ offspring aged 9–11 years (Figure 3), when disaggregated by gender, the vaccine uptake rate was 39.8% for daughters and 40.3% for sons at the national level. The northeast had the highest uptake for daughters (46.4%), while the islands had the highest uptake for sons (66.7%). The lowest uptake values were observed in the center for sons (29.8%) and in the islands for daughters (33.3%). In this age group, the Italian rates of individuals who were unaware of being the target of HPV vaccination were 42.5% for respondents with male offspring and 49.8% for respondents with female offspring.

Regarding respondents with children aged 12–17 (Figure 4), the national average uptake was 73.3% for daughters and 51.0% for sons. The highest uptake values were found in the south for daughters (78.5%) and in the northeast for sons (64.8%). Overall, the respondents who were unaware that their offspring were the target of HPV vaccination ranged from 30.4% for males to 16.9% for female offspring. 

Appendix A offers a comprehensive and detailed summary of the data presented in Figure 1, Figure 2, Figure 3 and Figure 4.

### 3.4. Worry about Getting an HPV Infection

The perception of HPV infection risk was analyzed by gender and geographical area, as shown in Appendix A. 

Among women aged 26 years and older (Appendix A), 44.2% expressed low concern about contracting HPV, while only 8.9% had a high perceived risk of HPV infection. These findings were consistent when examining the data by geographical area, with the majority of women indicating a low level of concern about becoming infected with HPV.

When analyzing the results by gender among respondents aged 18–25 years (Appendix A), 49.0% and 49.3% of males and females, respectively, reported having little concern about contracting HPV infection. Similarly, when examining the data by geographical area, a low-risk perception was the predominant response across all surveyed areas.

Regarding the perception of risk among parents of children aged 9–11 years (Appendix A), at the national level, there was a moderate perception of risk regarding their offspring getting infected with HPV, particularly for parents of daughters. However, for parents of sons, the perceived risk was, on average, lower (with 43.3% expressing moderate concern for daughters and 39.4% for sons). 

Among respondents with children between the ages of 12 and 17 years (Appendix A), the overall perception of the risk of their offspring contracting HPV was generally low. This applied to parents of both sons (46.4%) and daughters (39.5%), except in the southern regions where parental concern was moderate for daughters (with 40.9% of respondents expressing significant worry).

Appendix A offer a comprehensive and detailed summary of the data presented in Appendix A.

### 3.5. Safety of the HPV Vaccine

The findings on perceived safety of the HPV vaccine were similar across genders and geographical areas (Appendix A). This is evident when examining responses from women aged 26 years and older, parents of children aged 9–11, and parents of children aged 12–17. The prevailing perception at the national level is that the vaccine was considered as “quite safe”. 

### 3.6. Access to Healthcare Facilities, Vaccine Affordability and Payment 

Results regarding access to healthcare facilities providing vaccination are summarized in Appendix A. Upon disaggregating the data by gender, age group, and geographical area, it becomes evident that access to facilities was generally perceived as relatively easy in most cases. Specifically, 57.7% of women aged 26 and older found access to facilities quite easy (Appendix A). Among individuals between the ages of 18 and 25, 60.8% and 57.9% of male and female respondents, respectively, perceived access as quite easy (Appendix A). Among parents with children aged 9–11, 54.0% of parents with sons and 61.2% of parents with daughters described access to facilities as quite simple (Appendix A). Similarly, for children aged 12–17, access was considered quite simple by 61.7% of parents with sons and by 57.8% of parents with daughters (Appendix A). 

In terms of vaccine affordability, the respondents provided similar responses when examining the data by gender, age group, and geographical area (Appendix A). Approximately 50% of the sample (ranging from 49.3% to 58.2% across all target categories) regarded the vaccine as quite affordable.

Data on payment for HPV vaccination, when examined by gender, age group, and geographical area, are presented in Appendix A. Among female respondents aged 26 years and older (Appendix A), 64.6% reported that they received the vaccine for free. Among respondents aged 18–25 (Appendix A), 57.5% of males and 75.6% of females stated that they received the vaccine for free. 

### 3.7. Multivariable Regression Analysis

Older age, lack of awareness of vaccine priority status, and not having priority for HPV vaccination were significantly associated with an increased likelihood of both refusing and delaying vaccination (Appendix A). Furthermore, being unworried about HPV infection, perceiving HPV vaccines as unsafe, and having friends or relatives who are against vaccination were significantly associated with a higher probability of refusing vaccination. Delayed acceptance was significantly predicted by self-reported difficulties in accessing healthcare and mistakenly believing that one has higher priority for vaccination. Additionally, being female increased the likelihood of vaccine uptake (+18.3 percentage points compared to males). Region of residence, degree of urbanization, educational attainment, and perceived affordability of the vaccine did not show any relation to the study outcomes.

Analyzing potential interaction effects across covariates revealed that the impact of being unworried about HPV infection on vaccine refusal was much stronger among males (not worried: 50.6%; very worried: 18.0%; Δ = +32.6; 95% CI = 4.4 to 60.9) compared to females (not worried: 26.6%; very worried: 11.1%; Δ = +15.5; 95% CI = 10.1 to 20.9) (LR test = 29.5, *p*-value < 0.001). There was also evidence of a significant interaction (LR test = 33.3, *p*-value < 0.001) between safety concerns and gender, suggesting that such concerns had a greater impact on vaccine refusal among females (quite/very unsafe: 49.5%; very safe: 5.2%; Δ = +44.3; 95% CI = 38.7 to 50.0) compared to males (quite/very unsafe: 46.7%; very safe: 25.5%; Δ = +21.2; 95% CI = 5.8 to 36.7).

When analyzing vaccine uptake among children aged 9–17 years (Appendix A), we found that not being aware of the higher priority given to children for HPV vaccination was a significant predictor of both refusal and delayed acceptance. Furthermore, individuals with lower educational attainment, unworried about infection, and having safety concerns were significantly associated with a higher likelihood of refusing vaccination. Mothers and individuals living in southern Italy were significantly associated with a higher likelihood of delaying vaccination. We also found that female children and children aged 12 to 17 years had a significantly higher probability of vaccine uptake (females vs. males: +7.0 percentage points; 12–17 vs. 9–11 years: +13.1 percentage points). The degree of urbanization did not show any relation to the study outcomes, while perceiving the vaccine as “a little” or “not at all” affordable was significantly associated with increased uptake. Lastly, the analysis did not reveal any significant moderators among the covariates.

## 4. Discussion

This cross-sectional study examines the uptake of the HPV vaccine by stratifying the data based on age group, gender, and geographical area. The study enhances the existing data from the Italian Ministry of Health in terms of uptake [13], which mainly focuses on specific cohorts eligible for free vaccination. By investigating the social and behavioral factors related to vaccine refusal or delay, the OBVIOUS data reveal important insights into the underlying causes of low uptake rates among certain groups. These findings provide relevant information for a more nuanced understanding of the raw prevalence rates reported by the Italian Ministry of Health.

To better understand the HPV vaccine uptake rates, it is important to consider the context of the Italian vaccination program. Initially, the program offered free vaccination to 11-year-old girls and later expanded to include males born in the 2007 cohort (with some regions even including those born from 2004 onwards) [29]. Therefore, it is not surprising that the multivariable analysis identified female gender as a factor associated with a higher likelihood of accepting the vaccine. Additionally, the analysis revealed that older age was significantly linked to an increased probability of both refusing and delaying the vaccine.

The average HPV vaccine coverage among girls in Italy falls below the recommended threshold set by the National Vaccination Prevention Plan, which aims at a 95% coverage by the age of 12 [29]. None of the regions of Italy achieve a 95% coverage rate among the examined cohorts, highlighting significant variability across different geographic areas. This underscores the urgent need for targeted interventions in specific regions and emphasizes the importance of tailored approaches.

When it comes to adult women aged 26 years and older, the HPV vaccination coverage in Italy remains notably low, especially in certain geographical areas. However, it is worth noting that HPV vaccine uptake rates tend to be lower among adult women compared to adolescents. A recent study published in JAMA in 2021 found that, after weighting the data, approximately 42% of females received at least one dose of the HPV vaccine at any age. Furthermore, the percentage of females receiving the vaccine showed an increase from 32% in 2010 to 55% by 2018 [30].

There is a significant gender gap in coverage within the 18–25 age group, with women having a vaccination rate twice as much as men. This finding aligns with the adopted vaccination policies and is consistent with the latest ministry-level data as of December 31, 2021 [13,29]. It also confirms that Italy experiences a similar gender gap in HPV vaccine uptake among young adults compared to that found in studies conducted in other countries [30,31]. However, when comparing the gender gap between the 18–25 age group and the 12–17 age group, and even more so with the 9–11 age group, it is evident that over the years both males and females are being vaccinated almost equally. This trend is encouraging, as it holds promise for effectively achieving optimal coverage rates necessary for herd protection and health equity [32,33,34].

One major issue highlighted by this study is a substantial proportion of individuals not remembering whether they had received this specific vaccine. Furthermore, being unaware of being a target for vaccination, or whether their offspring are, emerged as strong drivers of refusal or delay in the multivariable analysis. These findings underscore the critical need for health literacy campaigns that not only increase knowledge about the HPV vaccine itself but also raise awareness of being eligible for free vaccination [35]. The discernment of knowledge lacunae in this investigation provides an avenue for Italian policymakers and public health practitioners to fortify precision-focused informational initiatives. Recent Italian pre- and post-intervention studies utilized questionnaires to assess the impact of educational sessions on HPV vaccination awareness among pre-adolescents. The results revealed a significant enhancement in awareness and a high vaccination rate among preadolescents. This highlights the pivotal role of health education programs and the involvement of physicians in counseling and recommending the vaccine [36,37].

Furthermore, despite the significant increase in uptake rates with age (especially among females), it is important to consider that HPV vaccination is most effective when administered early, ideally prior to the onset of sexual activity and potential exposure [38]. Therefore, it is necessary to target campaigns towards parents of young adolescents, emphasizing the importance of HPV early vaccination.

Vaccination awareness campaigns should simultaneously focus on increasing awareness of the risks associated with HPV infection and its related consequences, as our study revealed that 40–50% of respondents exhibit low concern regarding contracting the infection. However, the infection is associated with a significant burden of morbidity and mortality worldwide, including cervical cancer, genital warts, and other cancers [39].

Considering the observed interaction effects among covariates and the significant influence of low-risk perception on vaccine refusal and delay, it is advisable to not only implement campaigns aimed at increasing risk perception but also develop specific interventions targeting males. Our findings highlight that low-risk perception significantly contributes to vaccine refusal or delay among males, emphasizing the need for tailored campaigns to address this issue [40]. 

In terms of vaccine safety perception, our study yields more encouraging results compared to a 2021 cross-sectional study by Sonawane et al. published in JAMA, which reported a 79.9% increase in the proportion of parents refusing the HPV vaccine for their adolescents due to safety concerns [30]. In Italy, the HPV vaccine is currently regarded as “quite safe”. However, perception of the vaccine being unsafe is confirmed by our study as a significant predictor of vaccination delay or refusal. Therefore, it is crucial for strategies aimed at increasing HPV vaccine uptake to address individuals’ concerns regarding vaccine safety [35].

Public health professionals must also strive to enhance perceived accessibility to vaccinations [21], which in our study was considered as “quite easily accessible” for around 50–60% of respondents. Given that the acceptance of vaccines is impacted by pragmatic elements such as the accessibility of healthcare services, this study also scrutinized individuals’ inclinations in this context. We found that vaccination hubs were the facilities that received the highest satisfaction ratings as the preferred locations for receiving vaccines. Hence, in order to further improve vaccination accessibility, it would be helpful to diversify service delivery (e.g., hospitals, general practitioners’ offices, pharmacies, etc.) and consider an accurate vaccination site selection process [41].

### Limitations and Strengths

This study has some limitations that should be acknowledged. Firstly, due to its cross-sectional nature, causal relationships cannot be inferred from its findings. Secondly, given that the survey depended on self-reported responses in an online format, there exists the possibility of reporting bias. Moreover, there is a potential for selection bias, as only individuals capable of utilizing the online format were included in the study. Thirdly, the sample composition may not be fully representative of the general population as the survey attracted a larger proportion of individuals from lower socioeconomic classes, with 60.4% reporting economic struggles. This limitation may affect the generalizability of the results. Moreover, the economic status was evaluated relying on perceived income sufficiency rather than factual family income, as delving into income details might have influenced participants’ candor in completing the survey. Fourthly, the design of the survey included specific criteria for answering questions about HPV vaccination. If respondents met the demographic criteria related to the child population they answered on their children’s behalf. Otherwise, they provided information about their own vaccination status. This approach, aimed at brevity, may have introduced selection bias and should be considered when interpreting the data on adults. Lastly, no information about bivalent vs. quadrivalent vs. nonavalent HPV vaccination was collected in the questionnaire. Despite all these limitations, it is important to note that this study is the first of its kind and with this level of detailed and stratified results to provide data on HPV vaccination in Italy from a large national and representative sample.

## 5. Conclusions

The OBVIOUS study conducted provides crucial insights into the low uptake of the HPV vaccine in Italy. By comprehensively examining the factors influencing vaccine refusal or delay, particularly among specific age groups and geographic areas, this study offers valuable information for improving HPV vaccination coverage. The findings highlight several key intervention points that can inform program planning and guide policy makers’ decisions. These interventions include targeted approaches, tailored information campaigns, and heightened awareness of eligibility for free vaccination. Special attention should be given to promoting early vaccination and addressing the low-risk perception among males. Additionally, it is imperative to address concerns related to vaccine safety and enhance the perceived accessibility of vaccinations. By considering these factors, policy makers can effectively improve HPV vaccine uptake and mitigate the associated health risks of a disease which significantly impacts population morbidity and mortality.

## Figures and Tables

**Figure 1 vaccines-12-00187-f001:**
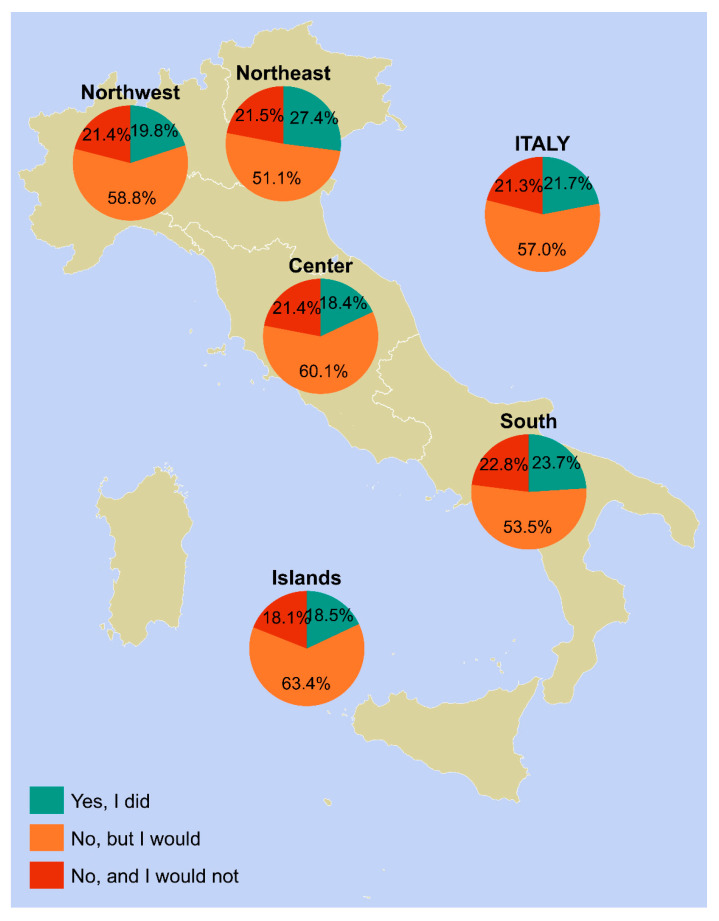
HPV vaccine uptake among female participants 26 years of age and older who answered on their own behalf (*n* = 2109). Notes: Females include non-binary people and participants who did not disclose their gender identity. Piedmont, Aosta Valley, Lombardy, and Liguria constitute northwestern Italy; Trentino-South Tyrol, Veneto, Friuli-Venezia Giulia, and Emilia-Romagna constitute northeastern Italy; Tuscany, Umbria, Marche, and Lazio constitute central Italy; Abruzzo, Molise, Campania, Apulia, Basilicata, and Calabria constitute southern Italy; and Sicily and Sardinia constitute insular Italy. Abbreviations: HPV, human papillomavirus; NUTS, Nomenclature of Territorial Units for Statistics.

**Figure 2 vaccines-12-00187-f002:**
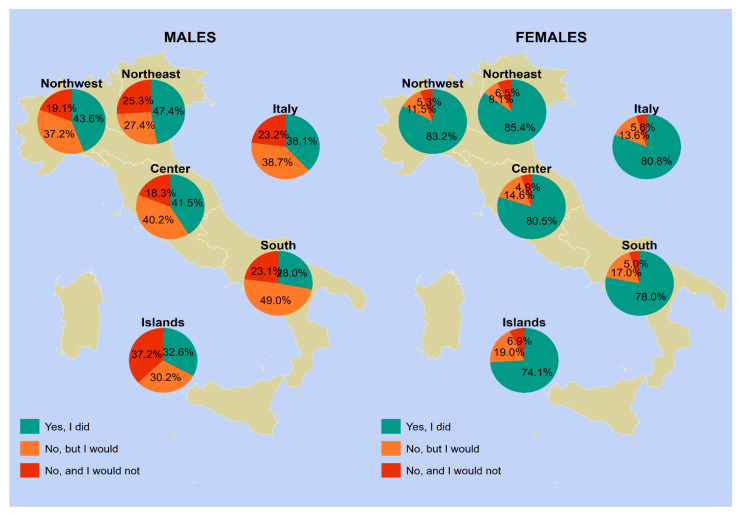
HPV vaccine uptake among male vs. female participants between 18 and 25 years of age who answered on their own behalf (*n* = 1051). Notes: Females include non-binary people and participants who did not disclose their gender identity. Piedmont, Aosta Valley, Lombardy, and Liguria constitute northwestern Italy; Trentino-South Tyrol, Veneto, Friuli-Venezia Giulia, and Emilia-Romagna constitute northeastern Italy; Tuscany, Umbria, Marche, and Lazio constitute central Italy; Abruzzo, Molise, Campania, Apulia, Basilicata, and Calabria constitute southern Italy; and Sicily and Sardinia constitute insular Italy. Abbreviations: HPV, human papillomavirus; NUTS, Nomenclature of Territorial Units for Statistics.

**Figure 3 vaccines-12-00187-f003:**
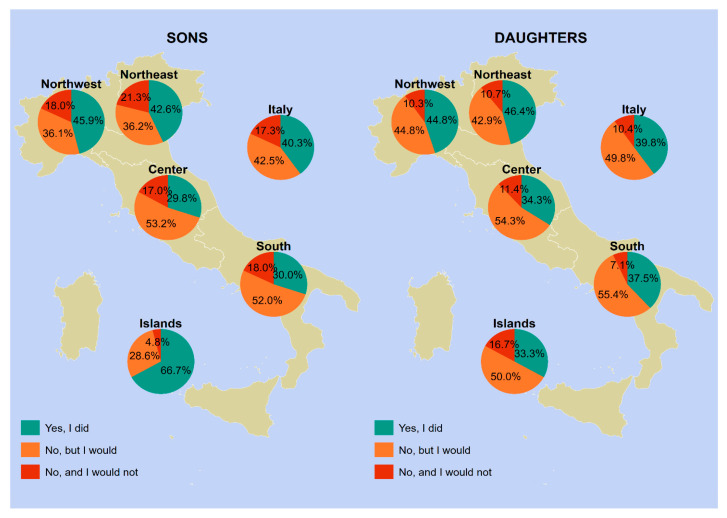
HPV vaccine uptake among male vs. female offspring between 9 and 11 years of age (*n* = 427). Notes: Information was provided by the parents of the children. Piedmont, Aosta Valley, Lombardy, and Liguria constitute northwestern Italy; Trentino-South Tyrol, Veneto, Friuli-Venezia Giulia, and Emilia-Romagna constitute northeastern Italy; Tuscany, Umbria, Marche, and Lazio constitute central Italy; Abruzzo, Molise, Campania, Apulia, Basilicata, and Calabria constitute southern Italy; and Sicily and Sardinia constitute insular Italy. Abbreviations: HPV, human papillomavirus; NUTS, Nomenclature of Territorial Units for Statistics.

**Figure 4 vaccines-12-00187-f004:**
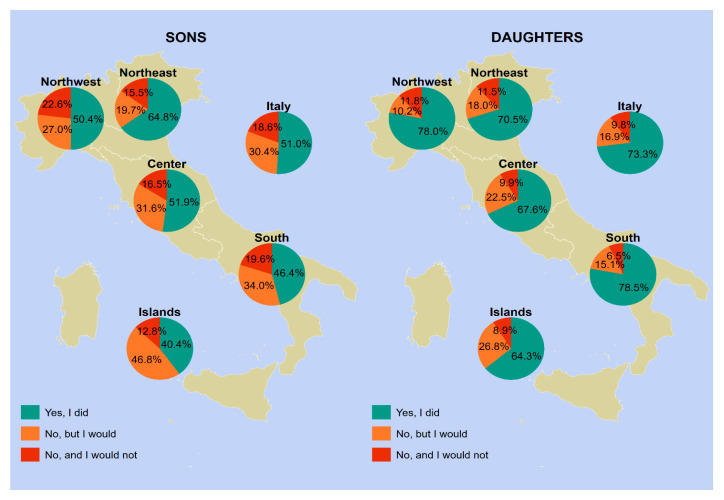
HPV vaccine uptake among male vs. female offspring between 12 and 17 years of age (*n* = 839). Notes: Information was provided by the parents of the children. Piedmont, Aosta Valley, Lombardy, and Liguria constitute northwestern Italy; Trentino-South Tyrol, Veneto, Friuli-Venezia Giulia, and Emilia-Romagna constitute northeastern Italy; Tuscany, Umbria, Marche, and Lazio constitute central Italy; Abruzzo, Molise, Campania, Apulia, Basilicata, and Calabria constitute southern Italy; and Sicily and Sardinia constitute insular Italy. Abbreviations: HPV, human papillomavirus; NUTS, Nomenclature of Territorial Units for Statistics.

**Table 1 vaccines-12-00187-t001:** Sociodemographic and clinical characteristics of the study participants who answered about their own HPV vaccine uptake, overall and by NUTS.

Characteristic	Italy	Northwestern Italy	Northeastern Italy	Central Italy	Southern Italy	Insular Italy
	(*n* = 3160)	(*n* = 776)	(*n* = 637)	(*n* = 634)	(*n* = 736)	(*n* = 377)
Gender						
Male	457 (14.5%)	94 (12.1%)	95 (14.9%)	82 (12.9%)	143 (19.4%)	43 (11.4%)
Female	2689 (85.1%)	679 (87.5%)	539 (84.6%)	551 (86.9%)	588 (79.9%)	332 (88.1%)
Non-binary	9 (0.3%)	1 (0.1%)	3 (0.5%)	1 (0.2%)	2 (0.3%)	2 (0.5%)
Prefer not to disclose	5 (0.2%)	2 (0.3%)	0 (0.0%)	0 (0.0%)	3 (0.4%)	0 (0.0%)
Age group, years						
18–25	1051 (33.3%)	225 (29.0%)	218 (34.2%)	205 (32.3%)	302 (41.0%)	101 (26.8%)
26–32 *	485 (15.3%)	113 (14.6%)	104 (16.3%)	101 (15.9%)	102 (13.9%)	65 (17.2%)
33–44 *	996 (31.5%)	271 (34.9%)	191 (30.0%)	201 (31.7%)	203 (27.6%)	130 (34.5%)
45–59 *	628 (19.9%)	167 (21.5%)	124 (19.5%)	127 (20.0%)	129 (17.5%)	81 (21.5%)
Place of residence degree of urbanization †						
City	1184 (37.5%)	353 (45.5%)	156 (24.5%)	259 (40.9%)	296 (40.2%)	120 (31.8%)
Town or suburb	1512 (47.8%)	335 (43.2%)	333 (52.3%)	294 (46.4%)	338 (45.9%)	212 (56.2%)
Rural area	464 (14.7%)	88 (11.3%)	148 (23.2%)	81 (12.8%)	102 (13.9%)	45 (11.9%)
Educational attainment						
Less than high school diploma	309 (9.8%)	77 (9.9%)	74 (11.6%)	64 (10.1%)	58 (7.9%)	36 (9.5%)
High school diploma	1903 (60.2%)	484 (62.4%)	358 (56.2%)	374 (59.0%)	443 (60.2%)	244 (64.7%)
Academic degree	734 (23.2%)	159 (20.5%)	164 (25.7%)	158 (24.9%)	187 (25.4%)	66 (17.5%)
Post-graduate/Doctorate degree	214 (6.8%)	56 (7.2%)	41 (6.4%)	38 (6.0%)	48 (6.5%)	31 (8.2%)
Household composition						
Alone	419 (13.3%)	118 (15.2%)	89 (14.0%)	97 (15.3%)	75 (10.2%)	40 (10.6%)
Couple	1361 (43.1%)	369 (47.6%)	286 (44.9%)	274 (43.2%)	262 (35.6%)	170 (45.1%)
With family of origin	1238 (39.2%)	250 (32.2%)	236 (37.0%)	235 (37.1%)	367 (49.9%)	150 (39.8%)
Other	142 (4.5%)	39 (5.0%)	26 (4.1%)	28 (4.4%)	32 (4.3%)	17 (4.5%)
Able to pay for things needed in life						
With great difficulty	402 (12.7%)	91 (11.7%)	77 (12.1%)	77 (12.1%)	99 (13.5%)	58 (15.4%)
With some difficulty	1481 (46.9%)	336 (43.3%)	285 (44.7%)	285 (45.0%)	373 (50.7%)	202 (53.6%)
Quite easily	1130 (35.8%)	300 (38.7%)	234 (36.7%)	247 (39.0%)	239 (32.5%)	110 (29.2%)
Easily	147 (4.7%)	49 (6.3%)	41 (6.4%)	25 (3.9%)	25 (3.4%)	7 (1.9%)
Daily problems due to physical, psychological, or sensory impairment						
Yes	242 (7.7%)	48 (6.2%)	54 (8.5%)	43 (6.8%)	70 (9.5%)	27 (7.2%)
No	2918 (92.3%)	728 (93.8%)	583 (91.5%)	591 (93.2%)	666 (90.5%)	350 (92.8%)
BMI ≥ 30 kg/m^2^						
Yes	287 (9.1%)	58 (7.5%)	62 (9.7%)	62 (9.8%)	71 (9.6%)	34 (9.0%)
No	2873 (90.9%)	718 (92.5%)	575 (90.3%)	572 (90.2%)	665 (90.4%)	343 (91.0%)
Pneumopathy						
Yes	177 (5.6%)	42 (5.4%)	41 (6.4%)	31 (4.9%)	39 (5.3%)	24 (6.4%)
No	2983 (94.4%)	734 (94.6%)	596 (93.6%)	603 (95.1%)	697 (94.7%)	353 (93.6%)
Cardiopathy						
Yes	121 (3.8%)	32 (4.1%)	32 (5.0%)	19 (3.0%)	29 (3.9%)	9 (2.4%)
No	3039 (96.2%)	744 (95.9%)	605 (95.0%)	615 (97.0%)	707 (96.1%)	368 (97.6%)
Diabetes						
Yes	135 (4.3%)	24 (3.1%)	40 (6.3%)	30 (4.7%)	28 (3.8%)	13 (3.4%)
No	3025 (95.7%)	752 (96.9%)	597 (93.7%)	604 (95.3%)	708 (96.2%)	364 (96.6%)

* Females only. † Graded using the Eurostat Degree of Urbanization (DEGURBA) classification system. Notes: Piedmont, Aosta Valley, Lombardy, and Liguria constitute northwestern Italy; Trentino-South Tyrol, Veneto, Friuli-Venezia Giulia, and Emilia-Romagna constitute northeastern Italy; Tuscany, Umbria, Marche, and Lazio constitute central Italy; Abruzzo, Molise, Campania, Apulia, Basilicata, and Calabria constitute southern Italy; and Sicily and Sardinia constitute insular Italy. Abbreviations: HPV, human papillomavirus; NUTS, Nomenclature of Territorial Units for Statistics; BMI, body mass index.

**Table 2 vaccines-12-00187-t002:** Sociodemographic characteristics of the study participants who answered about their youngest children’s HPV vaccine uptake, overall and by NUTS.

Characteristic	Italy	Northwestern Italy	Northeastern Italy	Central Italy	Southern Italy	Insular Italy
	(*n* = 1266)	(*n* = 383)	(*n* = 207)	(*n* = 232)	(*n* = 296)	(*n* = 148)
Gender						
Male	581 (45.9%)	180 (47.0%)	89 (43.0%)	120 (51.7%)	132 (44.6%)	60 (40.5%)
Female	684 (54.0%)	203 (53.0%)	118 (57.0%)	111 (47.8%)	164 (55.4%)	88 (59.5%)
Prefer not to disclose	1 (0.1%)	0 (0.0%)	0 (0.0%)	1 (0.4%)	0 (0.0%)	0 (0.0%)
Age group, years						
18–24	3 (0.2%)	1 (0.3%)	0 (0.0%)	1 (0.4%)	1 (0.3%)	0 (0.0%)
25–34	66 (5.2%)	13 (3.4%)	11 (5.3%)	10 (4.3%)	21 (7.1%)	11 (7.4%)
35–44	316 (25.0%)	94 (24.5%)	44 (21.3%)	55 (23.7%)	82 (27.7%)	41 (27.7%)
45–54	626 (49.4%)	196 (51.2%)	107 (51.7%)	115 (49.6%)	140 (47.3%)	68 (45.9%)
55–64	216 (17.1%)	66 (17.2%)	39 (18.8%)	47 (20.3%)	40 (13.5%)	24 (16.2%)
≥65	39 (3.1%)	13 (3.4%)	6 (2.9%)	4 (1.7%)	12 (4.1%)	4 (2.7%)
Place of residence degree of urbanization *						
City	527 (41.6%)	156 (40.7%)	74 (35.7%)	108 (46.6%)	138 (46.6%)	51 (34.5%)
Town or suburb	601 (47.5%)	190 (49.6%)	98 (47.3%)	100 (43.1%)	135 (45.6%)	78 (52.7%)
Rural area	138 (10.9%)	37 (9.7%)	35 (16.9%)	24 (10.3%)	23 (7.8%)	19 (12.8%)
Educational attainment						
Less than high school diploma	135 (10.7%)	48 (12.5%)	20 (9.7%)	24 (10.3%)	24 (8.1%)	19 (12.8%)
High school diploma	750 (59.2%)	230 (60.1%)	122 (58.9%)	140 (60.3%)	176 (59.5%)	82 (55.4%)
Academic degree	257 (20.3%)	71 (18.5%)	48 (23.2%)	44 (19.0%)	63 (21.3%)	31 (20.9%)
Post-graduate/Doctorate degree	124 (9.8%)	34 (8.9%)	17 (8.2%)	24 (10.3%)	33 (11.1%)	16 (10.8%)
Household composition						
Alone	45 (3.6%)	22 (5.7%)	6 (2.9%)	9 (3.9%)	5 (1.7%)	3 (2.0%)
Couple	997 (78.8%)	290 (75.7%)	169 (81.6%)	184 (79.3%)	237 (80.1%)	117 (79.1%)
With family of origin	91 (7.2%)	22 (5.7%)	8 (3.9%)	18 (7.8%)	31 (10.5%)	12 (8.1%)
Other	133 (10.5%)	49 (12.8%)	24 (11.6%)	21 (9.1%)	23 (7.8%)	16 (10.8%)
Able to pay for things needed in life						
With great difficulty	164 (13.0%)	48 (12.5%)	23 (11.1%)	32 (13.8%)	41 (13.9%)	20 (13.5%)
With some difficulty	653 (51.6%)	190 (49.6%)	111 (53.6%)	121 (52.2%)	157 (53.0%)	74 (50.0%)
Quite easily	395 (31.2%)	125 (32.6%)	67 (32.4%)	72 (31.0%)	85 (28.7%)	46 (31.1%)
Easily	54 (4.3%)	20 (5.2%)	6 (2.9%)	7 (3.0%)	13 (4.4%)	8 (5.4%)
Who takes charge of the child’s vaccinations						
Mostly myself	632 (49.9%)	190 (49.6%)	90 (43.5%)	105 (45.3%)	170 (57.4%)	77 (52.0%)
Mostly my partner	140 (11.1%)	35 (9.1%)	29 (14.0%)	28 (12.1%)	31 (10.5%)	17 (11.5%)
Equally myself and my partner	494 (39.0%)	158 (41.3%)	88 (42.5%)	99 (42.7%)	95 (32.1%)	54 (36.5%)

* Graded using the Eurostat Degree of Urbanization (DEGURBA) classification system. Notes: Piedmont, Aosta Valley, Lombardy, and Liguria constitute northwestern Italy; Trentino-South Tyrol, Veneto, Friuli-Venezia Giulia, and Emilia-Romagna constitute northeastern Italy; Tuscany, Umbria, Marche, and Lazio constitute central Italy; Abruzzo, Molise, Campania, Apulia, Basilicata, and Calabria constitute southern Italy; and Sicily and Sardinia constitute insular Italy. Abbreviations: HPV, human papillomavirus; NUTS, Nomenclature of Territorial Units for Statistics.

**Table 3 vaccines-12-00187-t003:** Demographic characteristics of the youngest children of the study participants who answered about their children’s HPV vaccine uptake, overall and by NUTS.

Characteristic	Italy	Northwestern Italy	Northeastern Italy	Central Italy	Southern Italy	Insular Italy
	(*n* = 1266)	(*n* = 383)	(*n* = 207)	(*n* = 232)	(*n* = 296)	(*n* = 148)
Gender						
Male	657 (51.9%)	198 (51.7%)	118 (57.0%)	126 (54.3%)	147 (49.7%)	68 (45.9%)
Female	609 (48.1%)	185 (48.3%)	89 (43.0%)	106 (45.7%)	149 (50.3%)	80 (54.1%)
Age group, years						
9–11	427 (33.7%)	119 (31.1%)	75 (36.2%)	82 (35.3%)	106 (35.8%)	45 (30.4%)
12–17	839 (66.3%)	264 (68.9%)	132 (63.8%)	150 (64.7%)	190 (64.2%)	103 (69.6%)

Notes: Piedmont, Aosta Valley, Lombardy, and Liguria constitute northwestern Italy; Trentino-South Tyrol, Veneto, Friuli-Venezia Giulia, and Emilia-Romagna constitute northeastern Italy; Tuscany, Umbria, Marche, and Lazio constitute central Italy; Abruzzo, Molise, Campania, Apulia, Basilicata, and Calabria constitute southern Italy; and Sicily and Sardinia constitute insular Italy. Abbreviations: HPV, human papillomavirus; NUTS, Nomenclature of Territorial Units for Statistics.

## Data Availability

All data are provided within the manuscript.

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
