# Peer review of "Human Papillomavirus (HPV) Vaccine Coverage and Confidence in Italy: A Nationwide Cross-Sectional Study, the OBVIOUS Project"

_vaccines, 2024, doi:10.3390/vaccines12020187_

Round 1

Reviewer 1 Report

Comments and Suggestions for Authors

This l study examines the uptake of the HPV vaccine by stratifying the data in Italy  based on age group, gender, and geographical area. The data is well presented and the results are conclusive. It also addresses the gap in 18-25 age group between men and women. 

It is not clear what kind of HPV vaccine ( quadri/nona valent) population the study population had.  The results can be sub characterized based on that information too. 

Also, according to new recommendations, women and men upto 45 age can get the vaccine. Discuss the data on basis of this fact.

Overall I recommend  speedy publication.

Author Response

First Reviewer

This l study examines the uptake of the HPV vaccine by stratifying the data in Italy  based on age group, gender, and geographical area. The data is well presented and the results are conclusive. It also addresses the gap in 18-25 age group between men and women. 

We express our gratitude to the reviewer for the time devoted to the manuscript's review and for the commendations on our work.

It is not clear what kind of HPV vaccine ( quadri/nona valent) population the study population had.  The results can be sub characterized based on that information too. 

We thank the reviewer for this comment, which allows us to further clarify this aspect. Indeed, respondents were not specifically asked about the type of vaccine they received, making it impossible to discuss disaggregated data on this matter. For this reason, we have now included the following sentence in lines 550–552: “Lastly, no information about bivalent vs. quadrivalent vs. nonavalent HPV vaccination was collected in the questionnaire”.

Also, according to new recommendations, women and men upto 45 age can get the vaccine. Discuss the data on basis of this fact.

We appreciate the reviewer for this comment. To our knowledge and in light of the recent Italian National Vaccination Prevention Plan, the vaccine continues to be offered to both boys and girls upon reaching the age of 11. Catch-up vaccination is recommended for women at least up to the age of 26, utilizing the opportunity of the first screening for cervical cancer prevention. For men, catch-up vaccination is recommended at least up to the age of 18, in cases where they have not been previously vaccinated or have not completed the vaccination cycle. Furthermore, there is an active and free offer of vaccination for other conditions as described in the manuscript (high-grade lesions due to HPV, MSM, etc.)(L75-L78, L85-L88).

Overall I recommend  speedy publication.

Reviewer 2 Report

Comments and Suggestions for Authors

Thank you for sharing your article assessing in a cross-sectional manner the nationwide coverage and confidence on HPV vaccine in Italy. The following comments may help to improve the article:

L91: Please the age of the 62.2% females and 49.9% males that received the HPV vaccine in 2020.

L93: I presume you mean the COVID pandemic?

L130: What kind of participants' background? Please clarify.

L134: Science is a broad field. Please be more specific.

L123: Did you perform a sample size calculation? What is your rationale for selecting 10,000 citizens? In L126-127 and L138-139 you bring up age stratification also in the context of gender. Did this have any implications on your sample size? L143-145, were those eligible to receive the vaccine free of charge also considered in your sample size stated?

General comment related to section 2.1: How did you obtain consent from participants despite the rather broad statement in L 634-635. Please provide more detail.

L154: Do you mean some kind if pilot-testing of the survey? What was your sample size chosen for this purpose and to whom was the testing addressed?

L163: "Necessary items" comprise what? 

L164: Was your target population affected by mental impairments? If so, what was the extent of impairment? This is just to assure that they were capable of competing the survey independently.

L166: Do you mean the anatomical or physical preferred location for vaccination? 

L160: Were the same 21 questions posed at females and males?

L187: Did you back translate the questionnaire into Italian to assure correct translation?

Table 1-3: "y" means years? Please explain any abbreviations used. L383: 12 and 17 years? Please, throughout, add the appropriate unit when reporting numbers.  

L539: So some kind of selection bias? 

L553-554: Is this sentence supposed to be in the manuscript? 

Comments on the Quality of English Language

Please see above. 

Author Response

Thank you for sharing your article assessing in a cross-sectional manner the nationwide coverage and confidence on HPV vaccine in Italy. The following comments may help to improve the article:

We express our gratitude to the reviewer for the time devoted to the revision of our manuscript and for the comments, which we believe have helped us improve and clarify certain aspects. Below is our point-by-point response.

L91: Please the age of the 62.2% females and 49.9% males that received the HPV vaccine in 2020.

As mentioned in lines 90–92, the percentages of vaccinated individuals pertain to individuals born in 2006. This means that, as of 2020, they were 14 years old.

L93: I presume you mean the COVID pandemic?

We thank the reviewer for this noticing this inaccuracy. The manuscript has been modified accordingly.

L130: What kind of participants' background? Please clarify.

We thank the reviewer for this comment. We meant “educational” background—the manuscript has been modified accordingly.

L134: Science is a broad field. Please be more specific.

We understand the reviewer's concern on this point. However, in the questionnaire administered in Italian, respondents were specifically asked about "science" in general. The survey was conducted in the period immediately following the COVID-19 pandemic, during which pseudo- or anti-scientific beliefs were common. Nevertheless, in this Methods section, we provided a comprehensive description of the questionnaire used by the OBVIOUS Project in its entirety. It is important to note that the final section, which included this specific question, has not been analyzed in this manuscript. This has been now reiterated in the second paragraph of the Materials and Methods section. The paper in which we addressed this topic can be found here: https://doi.org/10.3390/vaccines11040839.

L123: Did you perform a sample size calculation? What is your rationale for selecting 10,000 citizens?

We appreciate the reviewer for raising this point. As they correctly pointed out, conducting a formal power analysis would likely have resulted in a sample size of less than 10,000 statistical units. The rationale behind this decision is, frankly speaking, driven by budgetary constraints. Given our available resources, we aimed at maximizing the overall sample size to provide reasonably precise estimates of vaccine uptake for seasonal influenza, HPV, rotavirus, and other vaccines. Please note that subgroups of the overall sample were asked to answer each subsection of the OBVIOUS questionnaire. We have now clarified in the text (lines 197–200) that no formal power analysis was carried out. More precisely, the following sentences have been added to the paper: “No formal power analysis was conducted prior to data collection. Based on resource availability and the aim of collecting enough data to provide reasonably precise estimates originating from each section of the questionnaire, including further stratified analyses, we determined that a minimum size of 10,000 for the entire OBVIOUS sample.”

In L126-127 and L138-139 you bring up age stratification also in the context of gender. Did this have any implications on your sample size? L143-145, were those eligible to receive the vaccine free of charge also considered in your sample size stated?

We thank the reviewer for bringing this point to our attention. As stated in lines 124–128, Dynata’s allocation by gender, age group, and NUTS region was performed to ensure that the overall sample was representative of Italy’s demographics. As already addressed in the previous point, no formal power analysis was performed—rather, we maximized the overall sample size to provide reasonably precise estimates originating from each section of the questionnaire, including further stratified analyses. These analyses involve stratification by gender, age group, and NUTS, among others. With reference to the reviewer’s last point, those eligible to receive the HPV vaccine free of charge were included in the overall sample and accounted for a proportion of the sample more or less equal to that of Italy’s overall population. As expressed at the end of the Discussion section (fourth limitation), some degree of selection bias was introduced due to hierarchy rules that regulated the access to the survey as a parent or as a vaccine recipient.

General comment related to section 2.1: How did you obtain consent from participants despite the rather broad statement in L 634-635. Please provide more detail.

The informed consent was collected online by the panel provider for each participant before participating in the study, fully in compliance with the current legislation in Italy. Thanks to the reviewer’s comment, lines 627–628 have been rephrased as follows: “Online informed consent was obtained from all subjects prior to participating in the study”.

L154: Do you mean some kind if pilot-testing of the survey? What was your sample size chosen for this purpose and to whom was the testing addressed?

We thank the reviewer for enabling us to better explain this point. The testing phase aimed at evaluating the comprehension and response proficiency of individuals to the questions. This procedure involved 100 respondents and, through the analysis of the collected results, assisted in identifying potential sources of confusion, ambiguity, or misinterpretation. Ultimately, this process contributed to enhancing the reliability and validity of the questionnaire.

L163: "Necessary items" comprise what? 

We thank the reviewer for enabling us to better translate this expression. Respondents were indeed asked, in Italian, to inquire about any economic difficulties they might be experiencing.
Question #7 (Supplementary Material_Survey Tool): “With the financial resources available to you (from personal or family income), are you able to meet the needs of your current living conditions?”. The passage has been thus rephrased as “ability to pay living expenses”.

L164: Was your target population affected by mental impairments? If so, what was the extent of impairment? This is just to assure that they were capable of competing the survey independently.

We thank the reviewer for enabling us to better express this point. The correct translation of question #13 (Supplementary Material_Survey Tool) is as follows: “Due to a physical, psychological, or sensory disability, do you have difficulties completing daily tasks such as going to the doctor or buying groceries?”. The text and Table 2 have been revised accordingly. Essentially, we cannot directly infer the extent of mental impairment in our sample. However, because participation in the survey required proactive engagement, it is likely that cognitive or mental issues were virtually absent among study participants.

L166: Do you mean the anatomical or physical preferred location for vaccination? 

We thank the reviewer for enabling us to better specify this point. It concerns the locations where vaccinations were received and those where participants would prefer to receive them. The text has been revised accordingly.

L160: Were the same 21 questions posed at females and males?

Yes, they were. We have now specified in line 164 that information about pregnancy in late 2021 was exclusively asked to women.

L187: Did you back translate the questionnaire into Italian to assure correct translation?

Yes, we did. This has now been stated in lines 189–191 as follows: “Back translation was performed to check for the accuracy and soundness of the original Italian-to-English translation.”.

Table 1-3: "y" means years? Please explain any abbreviations used. L383: 12 and 17 years? Please, throughout, add the appropriate unit when reporting numbers.  

We thank the reviewer for noticing this inaccuracy. The manuscript has been modified accordingly.

L539: So some kind of selection bias? 

We thank the reviewer for enabling us to better specify this point. Indeed, not only did the survey depend on self-reported responses in an online format, potentially generating reporting bias, but this may have also led to a selection bias, as only those capable of utilizing the online format could participate.

L553-554: Is this sentence supposed to be in the manuscript? 

We thank the reviewer for noticing this inaccuracy. The manuscript has been amended.

Reviewer 3 Report

Comments and Suggestions for Authors

The study is very comprehensive and very good written. It should be accepted with a very few corrections.

Introduction:

Check the literature and the number of HPV types (more than 100; it would be around 450, Nat Rev Microbiol 2022 Feb;20(2):95-108).

Otherwise, Introduction part is very good written.

Material and methods:

Questionnaire is very well designed.

Results:

3.6. Access to healthcare facilities, vaccine affordability and payment:

The real numbers would be welcomed in this part particularly, and perhaps on other places all over the text.

Discussion:

Page 16: delete the part: “5. Conclusions 552 This section is not mandatory but can be added to the manuscript if the discussion is 553 unusually long or complex.”

Author Response

The study is very comprehensive and very good written. It should be accepted with a very few corrections.

We express our gratitude to the reviewer for the time devoted to the manuscript's review and for the commendations on our work.

Introduction: Check the literature and the number of HPV types (more than 100; it would be around 450, Nat Rev Microbiol 2022 Feb;20(2):95-108). Otherwise, Introduction part is very good written.

We express our gratitude for the reviewer's appreciation of the introduction and for the constructive comment that has enabled us to enhance the manuscript accordingly.

Material and methods: Questionnaire is very well designed.

Thank you once again for appreciating our work.

Results (3.6. Access to healthcare facilities, vaccine affordability and payment): The real numbers would be welcomed in this part particularly, and perhaps on other places all over the text.

We understand the reviewer's request to include absolute numbers in the descriptive part of the results. We made the decision to present only percentage values in the text due to the representativeness of the sample and to preserve readability. This choice would not have been made if the sample were not representative of the vaccine target Italian population. However, all the results described in the text are available (in both absolute values and percentages) in the tables presented in the manuscript and supplementary materials.

Discussion (Page 16): delete the part: “5. Conclusions 552 This section is not mandatory but can be added to the manuscript if the discussion is 553 unusually long or complex.”

We thank the reviewer for noticing this inaccuracy. The manuscript has been amended.

Round 2

Reviewer 2 Report

Comments and Suggestions for Authors

Thank you for addressing all my comments. The revised manuscript has much improved.